# Experimental Study on Extreme Hydrodynamic Loading on Pipelines. Part 1: Flow Hydrodynamics

**Behnaz Ghodoosipour [1,\*], Jacob Stolle [1], Ioan Nistor [1], Abdolmajid Mohammadian [1] and Nils Goseberg [2]**

[1] Department of Civil Engineering, University of Ottawa, Ottawa, ON, K1N 6N5, Canada
[2] Leichtweiß-Institute for Hydraulic Engineering and Water Resources, Technische Universität Braunschweig, 38106 Braunschweig, Germany
[\*] Correspondence: bghod068@uottawa.ca; Tel.: +1-613-562-5800

**Abstract:** Over the past two decades, extreme flood events generated by tsunamis or hurricanes have caused massive damage to nearshore infrastructures and coastal communities. Utility pipelines are part of such infrastructure and need to be protected against potential extreme hydrodynamic loading. Therefore, to address the uncertainties and parameters involved in extreme hydrodynamic loading on pipelines, a comprehensive experimental program was performed using an experimental facility which is capable of generating significant hydraulic forcing, such as dam-break waves. The study presented herein examines the dam-break flow characteristics and influence of the presence of pipelines on flow conditions. To simulate conditions of coastal flooding under tsunami-induced inundation, experiments were performed on both dry and wet bed conditions to assess the influence of different impoundment depths and still water levels on the hydrodynamic features.

**Keywords:** pipelines; extreme events; tsunami; dam-break wave; hydrodynamics

## 1. Introduction

### 1.1. Background

Recent devastating tsunami and storm surge events exposed the vulnerability of coastal communities to such extreme natural disasters. The number of people experiencing such catastrophic coastal flood events has been compounded by climate change and the ever-increasing urbanization of low-lying coastal areas all around the world [1]. This provided increased interest for research around the topic of extreme impacts on infrastructure. The need to study the hydrodynamic loading induced by such events and its effects on various structures is important. Coastal-induced inundation, due to tsunamis, hurricanes, and associated storm surges, can generate extreme turbulence, which impacts coastal areas and destroys infrastructures in their path. Moreover, sudden dam failure incidents can also cause similar impacts to vulnerable downstream infrastructures [2]. Understanding the dynamics of highly turbulent waves, and transient flows, as well as their interaction with structures, is complex and difficult to assess and quantify. This is among others, due to their highly non-linear and rapidly transient characteristics, and the common involvement of turbulent multi-phase processes [3]. Several researchers have conducted post-event forensic field surveys of the recent catastrophic events, such as the 2004 Indian Ocean and the 2011 Tōhoku Tsunami. Field survey results after the 2004 Indian Ocean Tsunami conducted in Khao Lak, Thailand, estimated coastal inundation heights between 4 to 7 m [4,5] and wave front celerities between 6 and 8 m/s [6]. Data recorded by the Japanese Port and Airport Research Institute (PARI) after the 2011

Tōhoku Tsunami in Japan showed inundation depths of up to 15 m in the city of Onagawa. During the same event, onshore inundation velocities of up to 10 to 13 m/s were also observed near Sendai Airport [7]. Fritz et al. [8] used video-processing of images filmed in Kasennuma Bay during the 2011 Tōhoku Tsunami and estimated inundation depths up to 9 m, flow velocities of up to 11 m/s and calculated Froude number values around 1. Field and numerical modelling of the 1993 Hokkaido-Nansei-Oki Tsunami revealed water depths of 5–15 m and flow velocities of 3–15 m/s. Results from such detailed surveys provide invaluable sources of hydrodynamics data for further analysis and comparison with experimental data or available analytical or empirical formula.

Several studies attempted to characterize the hydrodynamics of tsunami run-up on coastlines and its interaction with structures subjected to a solitary wave as a representative of the tsunami [9–12]. Aristodemo et al. [13] conducted a small-scale experimental study in a wave flume along with a numerical investigation using a smoothed-particle hydrodynamics (SPH) model. In their study, they investigated the induced loading from a solitary wave on a horizontal circular cylinder. Although solitary waves have been extensively used for tsunami-related studies, more in-depth studies have shown that such waves cannot represent real tsunamis properly, due to the discrepancies, such as difference in wave period, and height during the wave run-up [14]. Madsen et al. [15] concluded that the required evolutionary distance for an initial run-up into a solitary wave is well beyond the width of any oceanic dimension, concluding that solitary waves would not represent real tsunamis. Chan et al. [16], referred to the available data from the 2011 Tōhoku tsunami and concluded that the wavelength for real tsunami is significantly longer than for solitary waves generated in the laboratory.

Several studies tried to compare the different characteristics of solitary waves and a more realistic representative for tsunami-like waves, such as "bores". Leschka and Oumeraci [17] investigated the induced hydrodynamic forces from two different types of waves, namely solitary waves and bores, representing tsunamis, on three vertical cylinders with different arrangements numerically. They concluded that two different waves result in different flow hydrodynamics, i.e., wave height and flow velocity. Istrati et al. [18] investigated the different types of tsunamis, i.e., solitary waves and bores, and their effect on I-grider bridge with cross-frames. They characterized the induced forces from a bore as short-duration impulsive horizontal force at the time of bore impact that is followed by a smaller magnitude and longer duration forces. This was not observed in the case of solitary waves. Such findings emphasize the importance of choosing the correct type of tsunami representation for characterizing the induced forces on structures, especially for long waves. They also indicate that the unbroken solitary waves are not suitable wave models representing tsunamis.

Zhao et al. [19] studied the hydrodynamic properties of submarine pipelines under the impact of several widely used waves representing tsunamis, including solitary waves and N-waves with characteristics closer to a real tsunami. This study suggests that the hydrodynamic characteristics of the waves, such as water level, flow velocity, flow structure and induced forces in these methods are largely different. Moreover, the longer periods in tsunami-like waves causes a smoother water surface profile compared to solitary waves with shorter periods. As the wave passes the pipe, the size of the vortices generated downstream of the pipe in a solitary-wave is smaller than the vortices generated by tsunami-like waves. Moreover, by the time the wave passes the pipe, the wave height decreases faster for solitary waves as compared to tsunami-like waves, which in turn reduces the induced forces, as well as the duration of the acting force.

In summary, in this study, dam-break waves were used for studying the tsunami-like impact on pipes. Several researchers, i.e., References [20,21], have characterized such waves and stated that dam-break waves could represent real tsunamis.

In this study, where the assessment of tsunami-induced coastal floods on pipelines is addressed, a dam-break wave generated using a rapidly-opening swing gate was used to reproduce the highly turbulent flow conditions created during such extreme events. Stolle et al. [22] described details of the discussed dam-break waves. Several researchers characterized dam-break waves surging over a dry bed. Among them, Ritter [23], Henderson [24] and Chanson [25] developed solutions for the dam-break wave profile. The effect of bed condition (i.e., dry and wet bed) has also been the subject of

several studies. Chanson et al. [20] performed an experimental study on tsunami characteristics on wet and dry horizontal beds. They characterized wave momentum and wave front velocity at the beginning of the wave propagation in tsunamis and compared them to the classical dam-break waves. Wuthrich et al. [26] proposed a new method to generate bores over dry and wet bed conditions and investigated the influence of different parameters of the wave, such as the bore front celerity and the flow velocity profile. Moreover, studies, such as St. German et al. [27] and Douglas and Nistor [28] have investigated the effect of bed condition on tsunami characteristics numerically.

Several other researchers investigated the impact of dam-break and tsunami on structures. Nouri et al. [29], Al-Faesly et al. [30], Bremm et al. [31] and Foster et al. [32] evaluated the forces induced from tsunami-structure interaction under unsteady conditions. Other studies such as Wüthrich et al. [33,34] investigated the extreme hydrodynamic forces induced on buildings with various characteristics. Arnason [35] studied the interaction between an incident bore and a free-standing structure and focused on analyzing flow hydrodynamics in their presence. Goseberg et al. [36], studied different flow characteristics around vertical obstacles impacted by transient tsunami-like long waves. Studies, such as Araki and Deguchi [37], Mazinani et al. [38], and Chen et al. [39], investigated tsunami bore impact on coastal bridges.

An in-depth review of existing research in the context of extreme flow condition impact on different structures reveals a lack of investigations on the impact of transient tsunami-like waves on horizontal pipelines. The American Society of Civil Engineers (ASCE), through its ASCE7 Tsunami Loads and Effects Committee, has developed a new standard for tsunami impacts and loading [40]. Amongst the potential effects of such extreme events on infrastructure, this standard has emphasized the need to investigate tsunami loads on pipelines. The work presented herein is the first of a two-part paper which focuses for the first time, to the knowledge of the authors, on the impact of tsunami-like dam-break waves on submerged and above-ground horizontal pipelines in on wet and dry bed conditions. Part 1, focuses on the free-stream flow hydrodynamics and its alterations in the presence of a pipe impacted by such a flow. Part 2 [41] focuses on the hydrodynamic loading and associated force coefficients for horizontal pipelines located in coastal areas prone to tsunamis.

### 1.2. Objectives

The main goal of this study is to investigate the extreme hydrodynamic loading on pipelines under transient flows. The specific objectives of this first part of the two-part paper are to investigate and discuss the flow hydrodynamics of the dam-break waves and movement and impingement on a pipeline installed in its flow path. The following specific questions were examined in this study:

- What are the flow characteristics (time-history of the wave surface profile and flow velocity) for dam-break waves propagating over dry bed conditions for different wave heights?
- How are flow characteristics altered in the case of dam-break wave propagation over wet bed (still water on the flume bed, downstream of the impounding gate) and how are these characteristics changing when the dam-break wave height changes and/or when the still water depth of the wet bed varies?
- How do flow conditions get influenced by the presence of a horizontal cylindrical pipe immersed in the flow under both dry and wet bed conditions?

Results from this first paper will be further used to analyze and discuss the complex behavior of the hydrodynamic loading exerted on the pipe: In the companion paper [41].

## 2. Experimental Setup

### 2.1. Dam-Break Flume

A comprehensive experimental program was developed and conducted in the Dam-break Flume in the Hydraulics Laboratory at the University of Ottawa, Canada. Experiments were performed at a 1:25 length scale, under Froude similitude. The flume is 30.1 m long, 1.5 m wide, and 0.5 m high. A predetermined volume of water was impounded behind a rapidly-opening swing gate installed in

the flume to form an upstream reservoir with a length of 21.55 m and variable water depths. The dam-break waves were generated by the rapid opening of the swing gate. According to a previous study conducted in the same flume by Stolle et al. [22], the non-dimensional gate opening time, $T_0 = (t\sqrt{g/h})$ ($t$ being the gate opening time, $h$ being the impoundment depth), is dependent on $h$ with an approximately linear relationship as:

$$T_0 = 1.47\text{–}1.19\ h. \tag{1}$$

With the range of impoundment depths tested in this study, the non-dimensional gate opening time was in the range of $0.875 < T_0 < 1.113$. Therefore, the range of $T_0$, satisfied the criterion for acceptable non-dimensional gate opening time $T_0 < 1.41$ defined by Lauber and Hager [42]. A vertically-moving steel gate located at the downstream end of the flume ensured the control of the water level downstream of the swing gate and enabled, thus, the formation of wet bed conditions with different water depths. The tested pipe model was made of cold-rolled steel, 10 cm in diameter and 1.47 m in length, installed horizontally and transversally across the flume at $x$ = 5.6 m downstream of the gate. The pipe was designed to be perfectly rigid with very high natural frequency in order to avoid possible dynamic effects. Schematics of the flume, together with various experimental parameters, are shown in Figure 1.

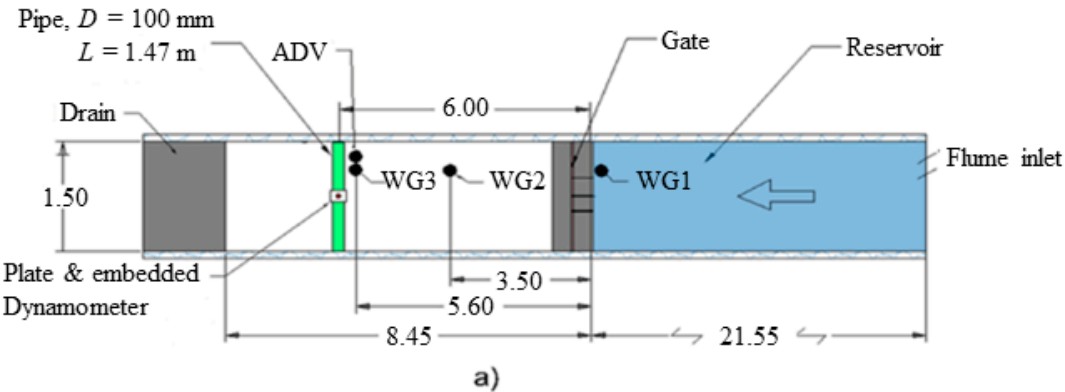

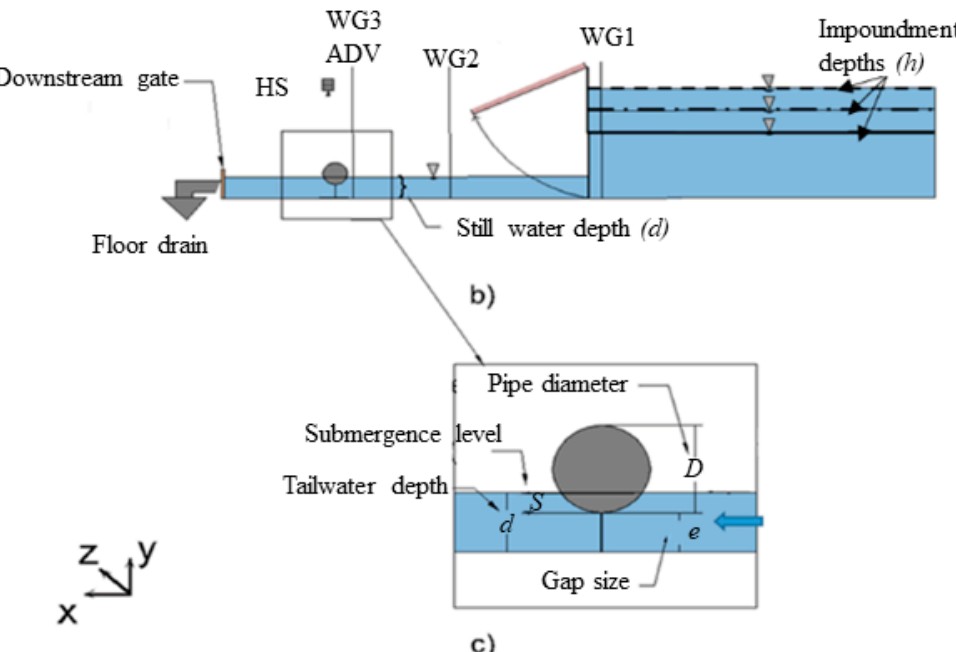

**Figure 1.** Flume and instrumentation sketch, (unless otherwise specified, all dimensions are in m). (**a**) Plan view, (**b**) side view and (**c**) close view with pipe and experimental parameters.

## 2.2. Instrumentation

Figure 2a,b show images of the flume together with the instruments and their locations. $x = 0$ was defined at the longitudinal position of the gate.

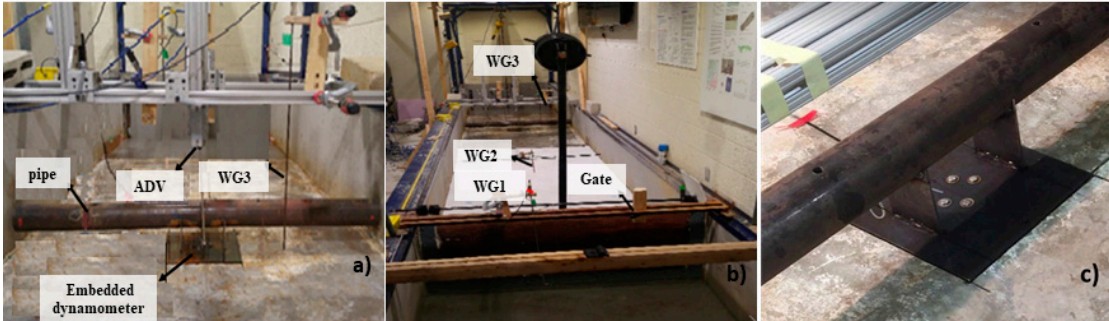

**Figure 2.** (**a**) Downstream view of pipe, dynamometer, ADV and wave gauge, (**b**) downstream view of flume and gate and (**c**) close view of pipe, supports and the base plate.

### 2.2.1. Wave Gauges

To record the time-history of the water level, three wave gauges (RBR WG-50, capacitance-type, ±0.002 m accuracy, RBR Global, Ottawa, ON, Canada) were installed at different locations along the flume. The first wave gauge (WG3) was installed upstream of the gate ($x = -0.04$ m) and was used to determine the opening time of the gate. The instant when the reservoir water level started to decrease, which was recorded by WG1, was used to synchronize the other measurement instruments The other two wave gauges were located at $x = 3.5$ m (WG2) and $x = 5.6$ m (WG1) downstream of the gate. The wave gauges sampling rate was 300 Hz. Wave gauges were calibrated by ensuring a linear gauge response with $R^2$ values greater than 0.99.

### 2.2.2. Acoustic Doppler Velocimeter (ADV)

A high-resolution acoustic Doppler velocimeter (ADV) (Vectrino, ±1 mm/s accuracy, 2.5 m/s measurement range, Nortek, Norway) was used for velocity measurements in the free stream flow. The velocity was used in the estimation of the drag and lift coefficients. The ADV was able to measure 3-D water velocities using coherent Doppler processing technology. In this study, a side looking ADV was used. The ADV's sampling rate was set to 200 Hz. The instrument was located at $x = 5.6$ m, 0.10 m upstream of the outer edge of the pipe. To derive the velocity profile, each experiment was repeated three times, and the ADV was moved vertically to different depths: (1) The highest water level; (2) the location where the center-axis of the pipe cross section was placed and 0.03 m above the flume bed. Non-uniformities in the cross-flow direction were assumed to be negligible. The water was seeded before each test using aluminum oxide powder with 27 micron particle size (400 mesh) to ensure adequate signal to noise ratio for ADV measurements.

### 2.2.3. Dynamometer

To record the time-history of the forces exerted on the pipe, a 6 degree of freedom (DOF) dynamometer (Interface- 6A68E, non-linearity, 0.04%, maximum capacity: *Fx = Fy* =10 kN, *Fz* = 20 kN, *Mx = My = Mz* = 500 Nm, Interface Inc., Scottsdale, AZ, USA*)* was used. This dynamometer was able to simultaneously measure the time-histories of the forces and moments along the three axes. The dynamometer was installed beneath the concrete flume floor by cutting the concrete flume floor, placing the device and re-embedding it, as shown in Figure 3a,b.

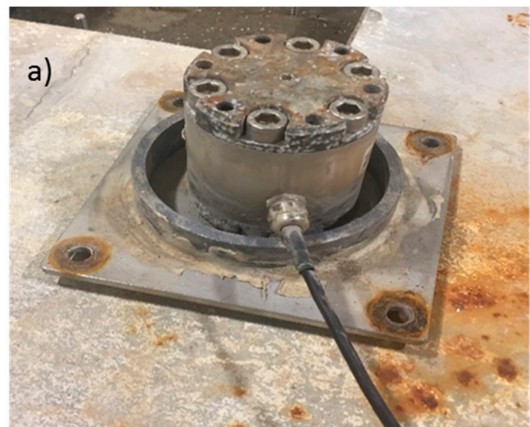 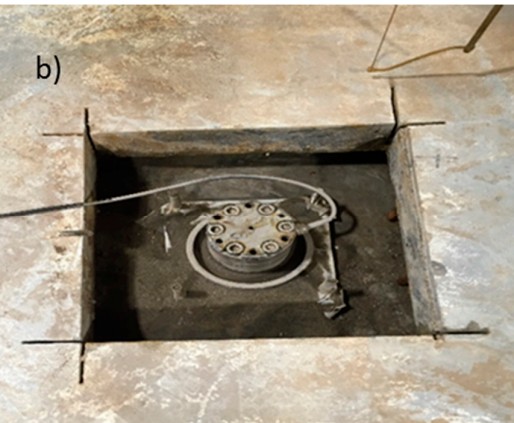

**Figure 3.** (**a**) Dynamometer (Interface-6A68E) and (**b**) dynamometer embedded in the flume floor.

A stiff steel plate was placed on top of the instrument, levelled with the flume. This plate transmitted the exerted force to the dynamometer through four bolts which rigidly fastened the instrument to the pipe model. The cable was securely placed in the flume floor recess, ensuring that no additional forces were transmitted on to the transducer. The pipe was attached to the plate using two very narrow, vertical plates, as shown in Figure 2a. The dynamometer was calibrated using the calibration chart provided by the manufacturer and re-zeroed before each test. The sampling rate for the dynamometer was set to 300 Hz. The amount of force and moment applied to the dynamometer was converted to voltage and recorded by the data acquisition system.

### 2.2.4. Data Acquisition System

Analog voltage signals from different instruments used in the experiments were converted to digital format and saved into data files using the QuantumX data acquisition system (MX840B, 8-channel universal amplifier and MX1601B with 16 individually configurable channels, HBM, Marlborough, MA, USA). All data were synchronized between the devices using a FireWire connection.

### 2.2.5. Camera

A camera (HS, Flare 2M360-CL, sampling rate 70 Hz, IO Industrial, London, ON, Canada) was directed towards the pipe from top to capture and analyze the bore impact with the pipe. A GoPro Hero4 Black (GoPro, San Mateo, CA, USA) was also installed 2 m upstream of the pipe and was used for observation purposes.

### 2.2.6. Cylindrical Pipe

A steel pipe, referred to as the cylindrical pipe, with an outer diameter of 100 mm, a wall thickness of 5 mm and a length of 1470 mm was used in the experiments. The pipe was connected to the upper plate bolted to the dynamometer using two brackets, 2 mm thick, made of steel, as shown in Figure 1a.

### 2.3. Experimental Test Program

Findings from this study were used in the companion paper (part 2) to characterize hydrodynamic forces exerted on pipelines, due to extreme flow events, modelled using a dam-break wave. A systematic and comprehensive experimental approach was conducted for this purpose. The most relevant parameters governing the problem at hand were varied during the experiments, namely: Reservoir depth ($h$), tailwater depth ($d$), lower edge of pipe distance to bed ($e$) to diameter ratio ($e/D$) and pipe level of submergence ($S$) to pipe diameter ratio ($S/D$). (Figure 1c). Table 1 shows the list of hydrodynamic tests in the absence of the pipe, while Table 2 illustrates the list of

experiments in the presence of a pipe with different experimental configurations. Each test was repeated three times to assess the repeatability of the results of each test. Head ratio ($d/h$) is defined as the ratio between the still water depth ($d$) and impoundment depth ($h$).

**Table 1.** List of hydrodynamic tests (no pipe).

| | Reservoir Depth $h$ (m) | Still Water Depth $d$ (m) | Head Ratio $d/h$ (-) |
|---|---|---|---|
| | | 0 | 0 |
| | | 0.03 | 0.1 |
| Hydrodynamic test (no pipe) | 0.3 | 0.06 | 0.2 |
| | | 0.08 | 0.26 |
| | | 0.12 | 0.4 |
| | | 0.17 | 0.56 |
| | | 0 | 0 |
| | | 0.03 | 0.075 |
| Hydrodynamic test (no pipe) | 0.4 | 0.06 | 0.15 |
| | | 0.08 | 0.2 |
| | | 0.12 | 0.3 |
| | | 0.17 | 0.425 |
| | | 0 | 0 |
| | | 0.03 | 0.06 |
| Hydrodynamic test (no pipe) | 0.5 | 0.06 | 0.12 |
| | | 0.08 | 0.16 |
| | | 0.12 | 0.24 |
| | | 0.17 | 0.34 |

**Table 2.** List of experimental configurations in the presence of a pipe.

| Gap Ratio $e/D$ (-) | Reservoir Depth $h$ (m) | Still Water Depth $d$ (m) | Head Ratio $d/h$ (-) | Level of Submergence Ratio $S/D$ (-) |
|---|---|---|---|---|
| | 0.3 | 0 | 0 | 0 |
| 0.3 | 0.40 | 0.03 | 0.1 | 0 |
| | 0.50 | 0.06 | 0.2 | 0.3 |
| | 0.30 | 0.08 | 0.26 | 0.5 |
| 0.6 | 0.40 | 0.12 | 0.4 | 1 |
| | 0.4 | 0 | 0 | 0 |
| | 0.30 | 0.03 | 0.075 | 0 |
| 0.8 | 0.40 | 0.06 | 0.15 | 0.3 |
| | 0.50 | 0.08 | 0.2 | 0.5 |

Test Repeatability (Water Level Time History)

Multiple tests with identical impoundment depths and initial still water depths were carried out to verify the repeatability of the tests. Figure 4 shows the water surface profile in dry and wet bed conditions measured by WG2 at $x$ = 3.5 m. Good agreement of water surface time-histories was achieved between multiple repetitions for both dry and wet bed conditions. Normalized standard deviations ($\sigma/h$) of less than 5% for wet bed and less than 4% for the dry bed were obtained.

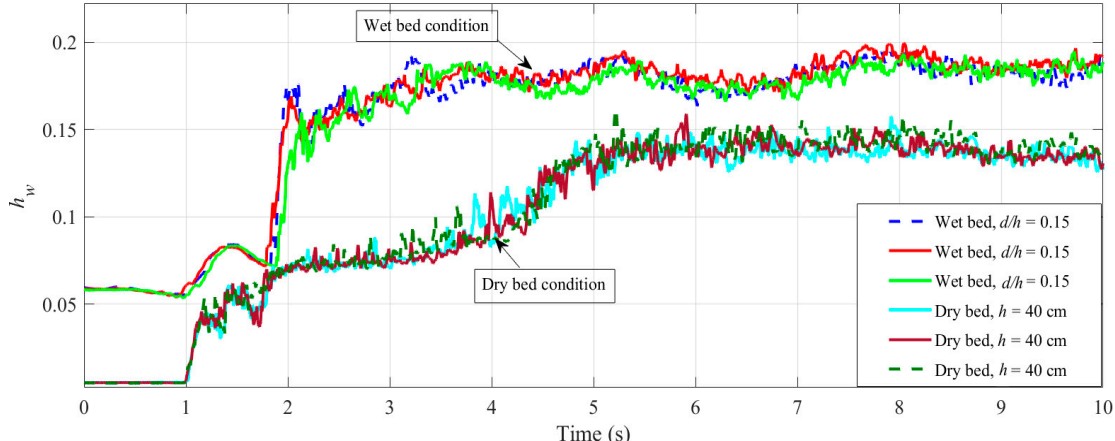

**Figure 4.** Repeatability of tests for water level time history (WG2), with various water depth (*h*) values of dry bed condition and *d/h* values of the wet bed condition.

## 3. Results and Discussion

### 3.1. Dry Bed Condition Hydrodynamics

3.1.1. Dry Bed Water Surface Profile

Figure 5 illustrates the measured water surface profile at the locations of the three wave gauges, shown in Figure 1. Figure 5a shows the normalized water level $h_w/h$ time-history at the location of the reservoir wave gauge (WG1), indicating a decrease in reservoir water depth in time. The gate opening time was used as the reference time in all three figures. Figure 5b,c illustrate the water surface profile for WG2 at *x* = 3.5 m and WG3 at *x* = 5.5 m (at the pipe location) downstream of the gate, respectively. Both Figure 5b,c illustrate the earlier wave arrival time for the dry bed surges generated using larger impoundment depth indicating a larger bore front celerity. The bore front celerity is discussed in more detail in Section 3.1.2. Comparison between water level magnitudes in Figure 5b,c also indicates a decrease in water level as the surge moves forward through the flume. The average non-dimensional water level ($\frac{h_w}{h}$) was decreased by 32% at WG3 compared to WG2. This can be explained with energy losses, due to bed friction in the case of flow on dry bed condition.

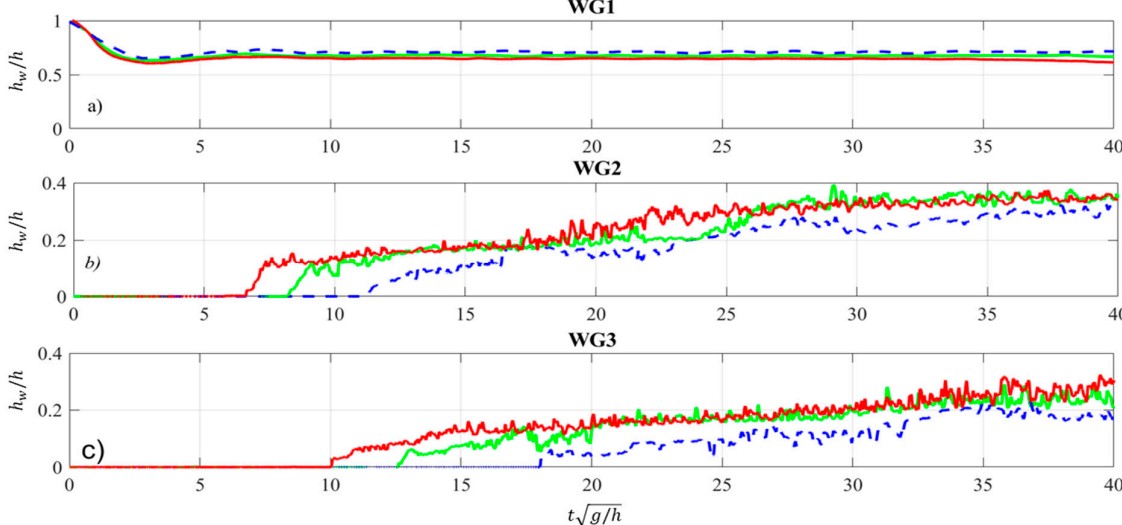

**Figure 5.** Dry bed surge time-history of the water surface profile. *h* = 50, 40, 30 cm measured at location of the (**a**) reservoir wave gauge (WG1), (**b**) *x* = 3.5 m (WG2) and (**c**) *x*= 5.5 m (WG3).

Figure 6 compares experimental results for the water surface profiles for dam-break flow in dry bed conditions, obtained from multiple tests with identical gate opening conditions and wave gauge locations along the flume, with the analytical solution of the Saint-Venant equations for a horizontal, frictionless surface given by Ritter [23] as:

$$\frac{h_w}{h} = \frac{1}{9}\left(2 - \frac{x}{ht\sqrt{\frac{g}{h}}}\right)^2 \tag{2}$$

In Figure 6, water surface profiles are plotted versus the dimensionless time, $t\sqrt{g/h}$. Experimental and Ritter's theoretical solution were compared at WG2 (x = 3.5 m) for three different impoundment depths h = 30, 40, and 50 cm. Figure 6 shows that, initially $(0 < t\sqrt{g/h} < 2)$, the experimental results do not accurately match the Ritter [23] solution. This observed discrepancy is due to the fact that the surface roughness of the flume bed in the Ritter [23] solution is ignored (fully smooth bed). For the case of the dry bed conditions and at the beginning of a dam-break wave surge, roughness plays a significant role as there is direct contact between the bore front and flume surface. The experimental results agree with a study conducted by Lauber and Hager [42] where the bed roughness was shown to have a significant effect close to the wave fronts. They further concluded what other studies, i.e., Wüthrich et al. [26] also found, namely Ritter's solution does not accurately represent the dam-break flow at the vicinity of the wave front because of the bed roughness.

### 3.1.2. Dry Bed Bore Front Celerity

The average front celerity of the propagating wave was estimated using the following ratio:

$$U = \frac{\Delta x}{\Delta t}, \tag{3}$$

where $U$ is the bore front celerity in m/s. $\Delta x$ is the distance between two wave gauges downstream of the gate, WG2 and WG3, and is equal to 2.1 m. The surge travel time is shown with $\Delta t$, as the time taken by the bore to travel between the two wave gauge positions.

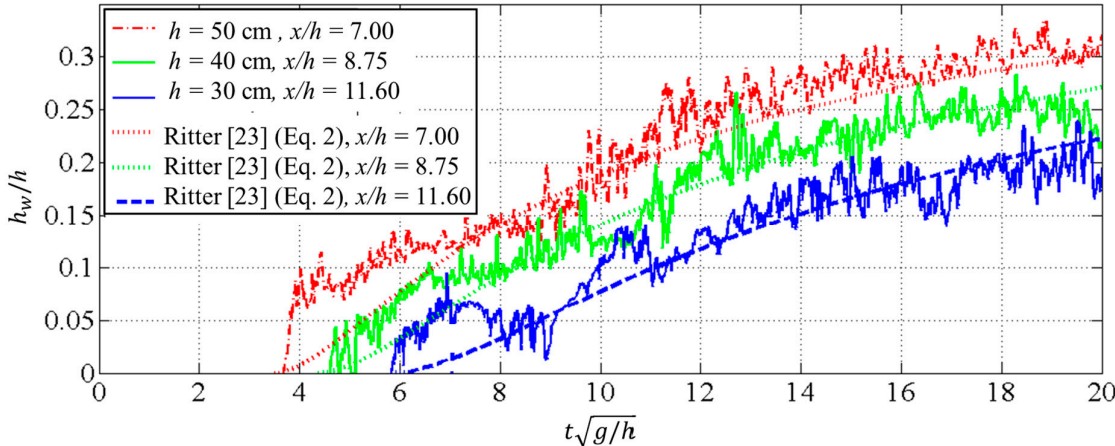

**Figure 6.** Non-dimensional dry bed condition water time-history surface profile: Comparison with Ritter's (1892) solution. $\frac{h_w}{h}$ versus non-dimensional time $t\sqrt{g/h}$ with $\frac{x}{h}$ = 7, 8.75, 11.6.

Several previous studies have estimated the bore front celerity *(U)* relative to impoundment depth *(h)* using:

$$U = \propto \sqrt{gh}, \tag{4}$$

where $\alpha$ is a constant with various values reported in the literature. The constant value depends on the flume hydraulic radius and roughness coefficient. Wüthrich et al. [26] suggest $\alpha = 1.25$, while Matsutomi and Okatamo [4] suggest $\alpha = 1.1$. Figure 7 shows results from the current study, together with the previous studies mentioned above. This study suggests $\alpha = 1.2$ as the constant in Equation (3).

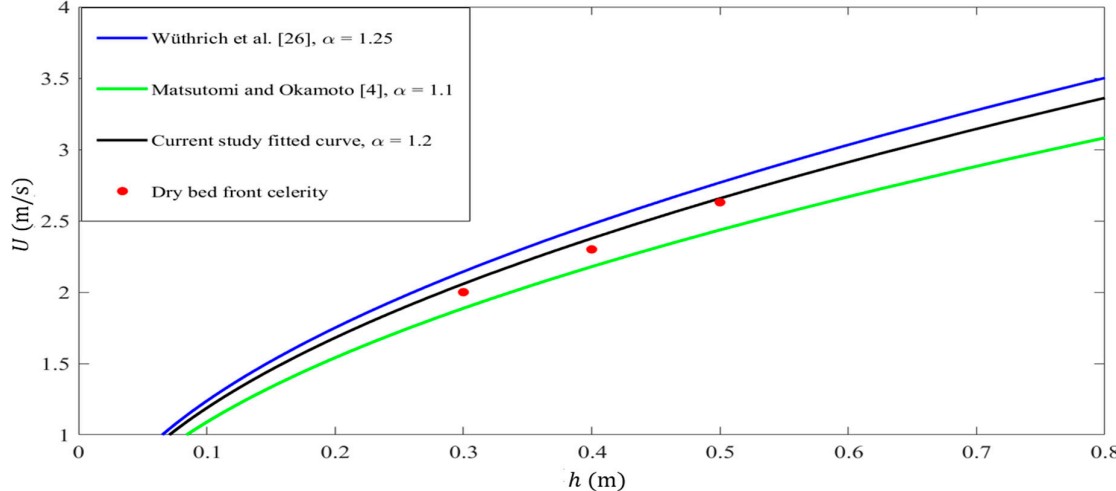

**Figure 7.** Comparison between the front celerity for dry bed in this study and previous studies.

### 3.1.3. Dry Bed Flow Velocity, Froude Number and Momentum Flux

Flow characteristics at the pipe location were studied to analyze the exerted forces on the pipeline. Figure 8 illustrates the dry bed surge characteristics as water surface profile (WG3), flow velocity, Froude number and momentum flux at the location of the pipe, respectively.

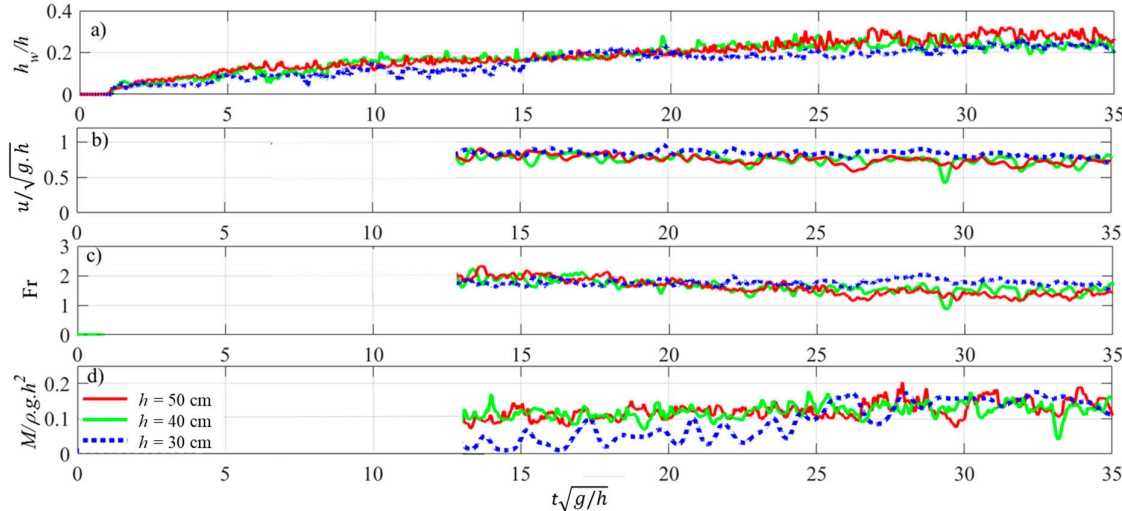

**Figure 8.** Dry bed surge characteristics at $x = 5.5$ for $h = 30, 40, 50$ cm. (**a**) water surface time-history profile (WG3), (**b**) flow velocity time-history, (**c**) Froude number time-history and (**d**) momentum flux time-history.

The reference time for all the cases is the wave arrival time at WG3. There is a delay in the velocity measurements using the ADV, due to the air entrainment close to the wave arrival time which corresponds to the zone with no data in the first few seconds in Figure 8b–d. Calculated Froude numbers (Fr $= \frac{u}{\sqrt{g h_w}}$) for dry bed conditions and for different reservoir impoundment depths are

shown in Figure 8c. It should be noted that for the impoundment depth of $h$ = 30 cm, the water level increases and the flow velocity decreases more gradually compared to the cases with $h$ = 40 cm and $h$ = 50 cm. The dry bed surge was supercritical, Fr > 1, throughout the studied time frame for all three impoundment depths, as shown in Figure 8c. The Froude number remains almost constant in the case of $h$ = 30 cm whereas, it gradually decreases for $h$ = 40 cm and $h$ = 50 cm.

The momentum flux per unit width $M$ is an important factor directly affecting the hydrodynamic loading on structures.

$$M = \rho h_w u^2, \tag{5}$$

where $h_w$, is the water level and $u$ is the depth-averaged flow velocity. Figure 8d shows the non-dimensional computed momentum flux as $\frac{M}{\rho g h^2}$, for dry bed and different impoundment depths. Figure 8d illustrates that the non-dimensional momentum is smaller for $h$ = 30 cm at the beginning of the surge. This could be explained by considerably smaller flow velocity, as well as small water depth in the case of $h$ = 30 cm. Smaller momentum flux results in smaller induced drag force as is discussed in the companion paper.

### 3.2. Wet Bed Condition Hydrodynamics

### 3.2.1. Wet Bed Water Surface Profile

Figure 9 illustrates the water surface profile for wet bed condition with impoundment depth of $h$ = 40 cm and different still water levels *(d)* at the location of three wave gauges WG1 (a), WG2 (b) and WG3 (c). The reference time in the figure was the gate opening time. The figure shows earlier arrival time for the cases with smaller still water depth (wet bed condition) which indicates a larger bore front celerity in such cases. The values recorded by WG2 and WG3, shown in Figure 9b,c, did not exhibit any change in the water surface profiles. Stoker [43] called the region where the water level remains constant "zone of constant state".

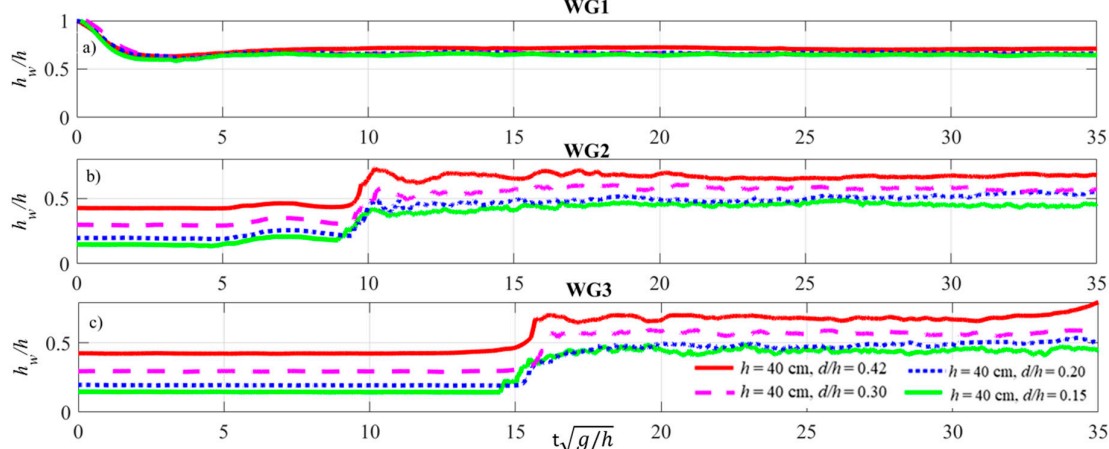

**Figure 9.** Wet bed bore water surface time history for $h$ = 40 cm, different still water levels, $d$ = 6, 8, 12, 17 cm, measured at location of (**a**) reservoir wave gauge (WG1), (**b**) $x$ = 3.5 m (WG2) and (**c**) $x$ = 5.5 m (WG3).

Figure 10 shows a comparison between dry bed and wet bed condition water surface time-histories. The data shows a steeper bore front and more abrupt water level rise in the case of the bore propagating over wet bed when compared to dry bed. Other researchers, i.e., Nouri et al. [29] and Wüthrich et al. [26], also found a similar behavior. According to Wüthrich et al. [26] the behavior of a bore propagating on the wet bed at the wave front, is similar to a turbulent and highly aerated hydraulic jump which causes a more abrupt water level rise compared to wave propagating over dry bed condition.

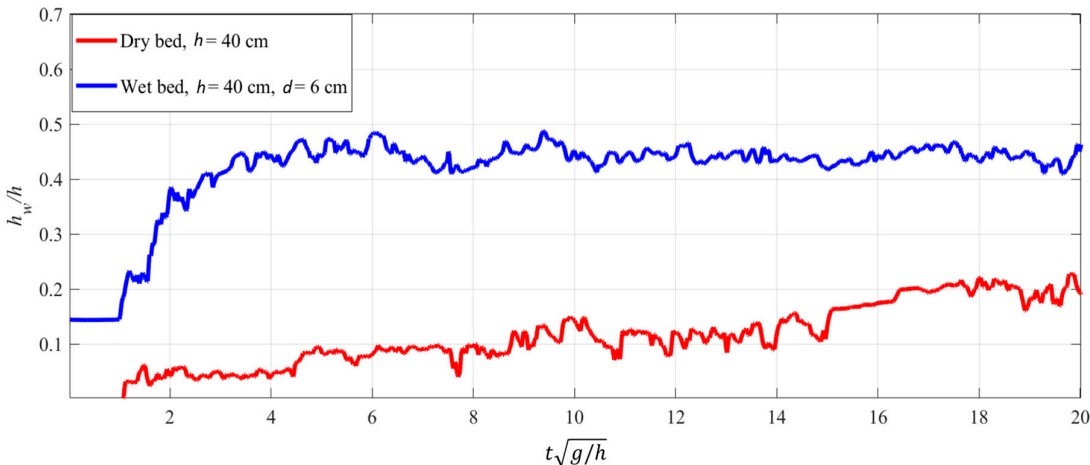

**Figure 10.** Comparison between dry bed and wet bed condition normalized water depth time-history.

### 3.2.2. Wet Bed Bore Front Celerity

The bore front celerity was calculated using Equation (3) for the case of wet bed condition for different reservoir impoundment and downstream still water depths. Figure 11 illustrates the dimensionless bore front celerity versus head ratio *(d/h)* obtained from this study together with Chanson [25] empirical solution for bore front celerity in a horizontal channel initially filled with water as:

$$\frac{U}{\sqrt{gd}} = \frac{0.6345 + 0.3286(\frac{d}{h})^{0.65167}}{0.00251 + (\frac{d}{h})^{0.65167}}. \tag{6}$$

Results show good agreement between experimental data and the solution of Chanson [25], which depicts the validity of the empirical solution proposed by this author.

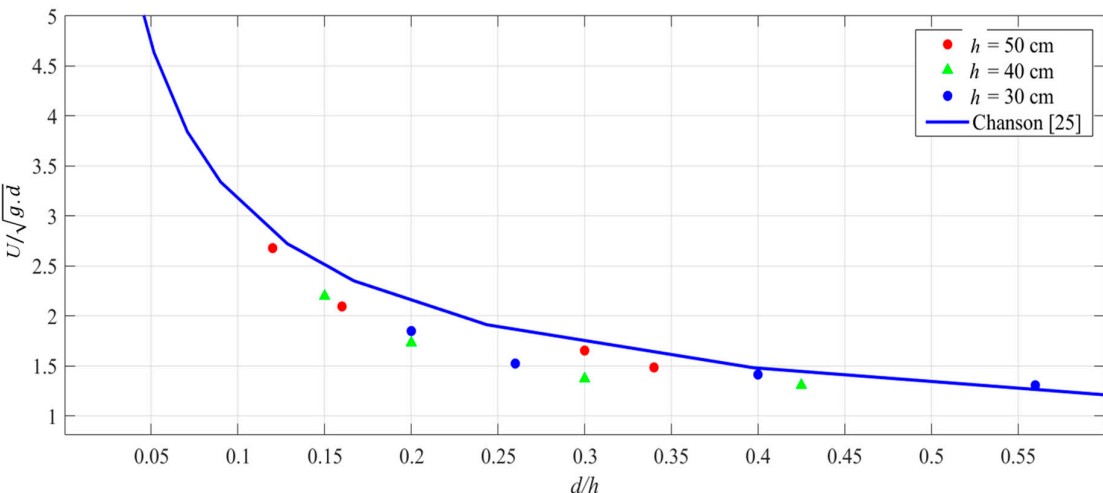

**Figure 11.** Bore front celerity *d/h*. The solid line shows Chanson [25] solution (Eq. 6), while the points show experimental data.

### 3.2.3. Wet Bed Flow Velocity, Froude Number and Momentum Flux

Figure 12 depicts the wet bed bore flow velocity, the computed Froude numbers and momentum flux in the case of wet bed condition, a constant impoundment depth of *h* = 40 cm, and different still

water depths *(d)*. The reference time for all the cases is the wave arrival time at WG3. Due to the air entrainment close to the wave arrival time, the flow velocity data at the beginning of the bore propagation were considered invalid and were eliminated from Figure 12b–d. Results show a noticeable decrease in flow velocity (Figure 12b) and the estimated Froude number (Figure 12c), with an increase in the still water depth or *d/h* ratio.

This is because such waves were generated using a smaller pressure head (the small difference between the upstream impoundment depth and the downstream still water depth) which resulted in slower flow velocities and lower Froude numbers. Results from all the three tested impoundment depths, i.e., *h* = 30, 40, 50 cm, show that for *d/h* 0.3 the flow was subcritical, while for *d/h* ≤ 0.2 the flow was supercritical. Head ratio values around 0.2 resulted in critical flow regime with Froude number values fluctuating around one. Figure 12d shows the calculated non-dimensional momentum flux($\frac{M}{\rho g h^2}$) values in the case of wet bed condition for impoundment depth of *h* = 40 cm.

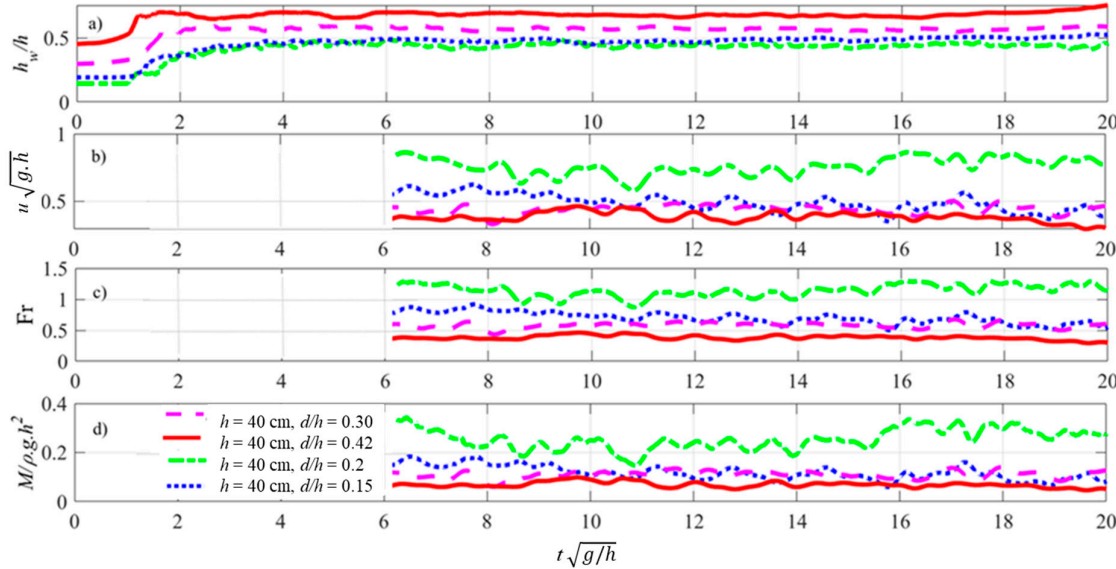

**Figure 12.** Wet bed bore characteristics at *x* = 6.5 for *h* = 40 cm, different line types show different *d/h* values. (**a**) Water surface time-history profile (WG3), (**b**) flow velocity time-history, (**c**) Froude number time-history and (**d**) momentum flux time-history.

Results show that the momentum flux decreases as the value of *d/h* increases. Smaller flow velocities in the bores generated using a smaller head (larger *d/h*) resulted in smaller momentum values. The same trend was observed for the case with *h* = 30 cm and *h* = 50 cm.

Chanson [25] presented a solution for a dam-break wave moving over a frictionless horizontal channel initially filled with water. The basic flow equations in wet bed conditions are the characteristic system of equations for simple waves as forward and backward characteristics. The forward characteristic in wet bed condition satisfies:

$$V_2 + 2\sqrt{gh_w} = V_0 + 2\sqrt{gh}, \tag{7}$$

where *h* is the reservoir depth and $V_2$ and $h_2$ are the flow velocity (m/s) and bore depth (m) immediately behind the positive surge. The quantity $V_0$ is the initial reservoir velocity (m/s) equal to zero in the current experiments. Chanson [25] solved this equation together with the continuity and momentum equations graphically. Figure 13 compares Chanson [25] graphical solution and $V_2$ values measured in current experimental work in a test with *d/h* = 0.2 (*h* = 40 cm, *d* = 8 cm). Results showed good agreement between the two studies.

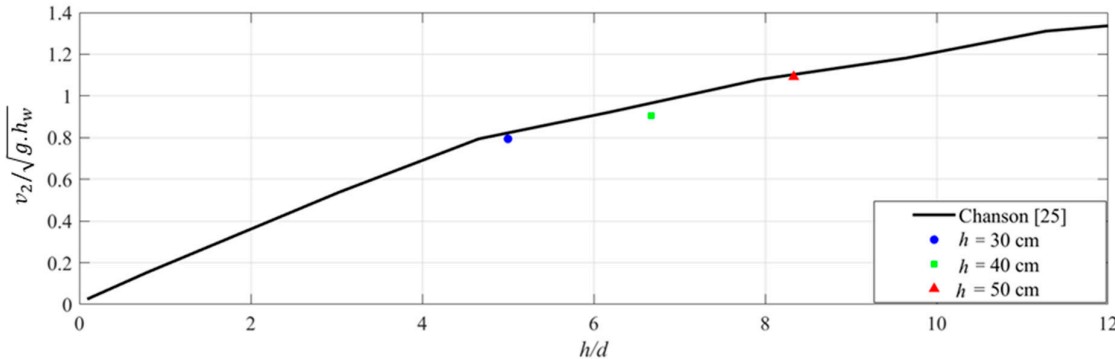

**Figure 13.** Tests with the wet bed condition flow velocity compared to Chanson's [25] graphical solution.

### 3.3. Changes in Hydrodynamic Conditions Due to the Presence of the Pipe

### 3.3.1. Dry Bed Condition

Influence of Pipe Gap Ratio (*e/D*) in Dry Bed Condition

Experimental results in the presence of the pipe showed a considerable change in flow hydrodynamic characteristics, i.e., in the water level and flow velocity. Figure 14 illustrates the alterations in the flow hydrodynamics in dry bed condition in the presence of the pipe with three different gap ratios (*e/D*) compared to the flow in the absence of the pipe.

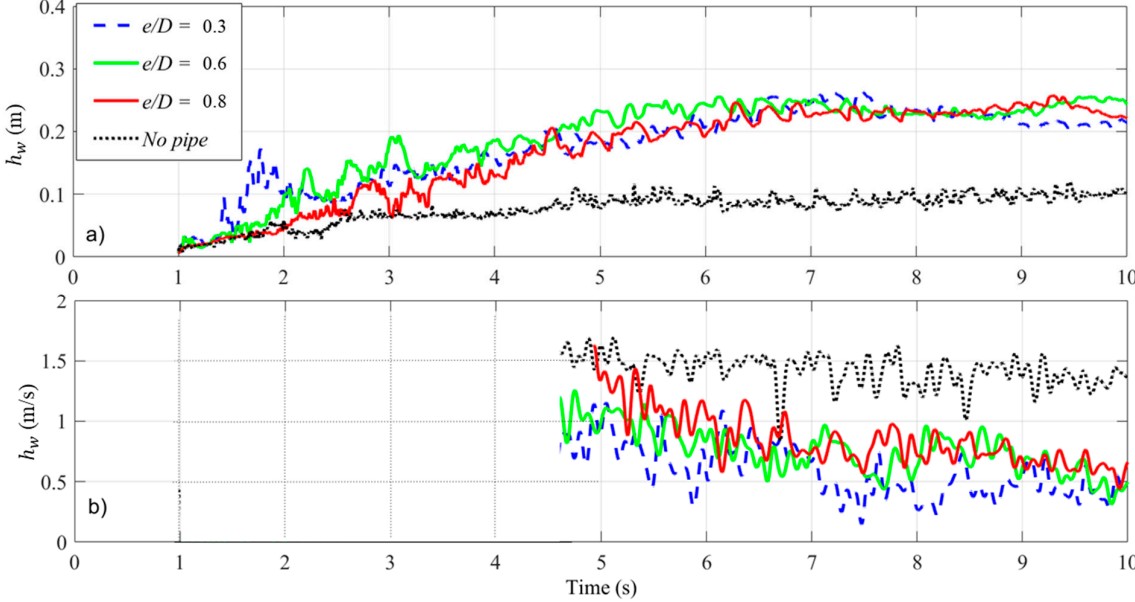

**Figure 14.** Effect of pipe existence in flow hydrodynamics for different *e/D* values, dry bed condition and *h* = 40 cm. (**a**) Water level time history and (**b**) flow velocity time-history.

The figure also shows that in the case of the smallest gap ratio (*e/D* = 0.3), water reached to the pipe surface faster than other cases and caused abrupt water level rise, as shown in Figure 14 (dashed line at *t* = 1.8 s). The water level rise at the pipe location was due to the flow being blocked by the pipe at the time of wave impact. Results of the flow velocities presented in Figure 14b show that the flow velocity reduced in the presence of the pipe, due to flow blockage by the latter. The flow velocity decreased to smaller values after the partial blockage by the pipe. This happened faster as the gap

ratio *e/D* decreased from 0.8 to 0.3, since the water reached the lower edge of the pipe and was blocking the incoming flow sooner. At lower gap ratios, i.e., *e/D* = 0.3, the pipe also got fully submerged earlier which reduced the flow turbulence caused by wave run-up and resulted in the flow velocities to be lower compared to the cases with larger *e/D* (Figure 14b).

Influence of Impoundment Depth

Figure 15 illustrates the water level rise at the time of bore impacting the pipe. The water level rise is more pronounced for the bore produced by the *h* = 50 cm impoundment depth (Figure 15a) and smallest for that generated by the *h* = 30 cm impoundment depth (Figure 15c). Such abrupt water level rise in larger impoundment depth results in more pronounced impulse force in such cases. Force time-histories are discussed in more detail in the companion paper [41].

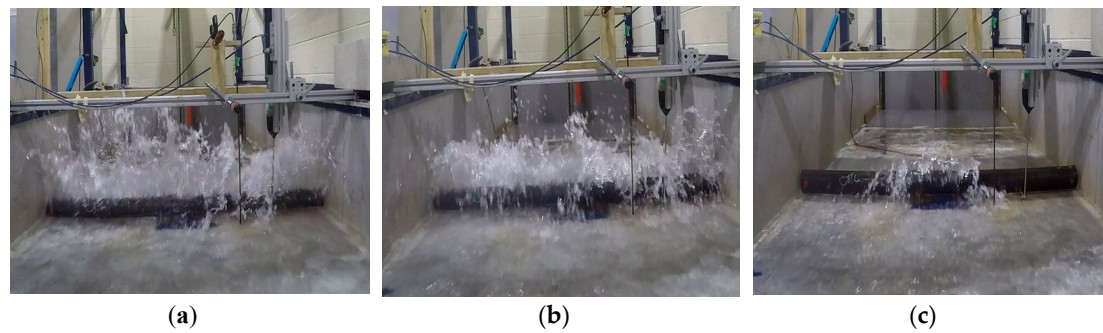

|       (a)       |       (b)       |       (c)       |

**Figure 15.** Water level rise at the time of bore impact generated by impoundment depths of (**a**) *h* = 50 cm, (**b**) *h* = 40 cm and (**c**) *h* = 30 cm.

3.3.2. Wet Bed Condition

Influences of Changing Still Water Depth (*d*) and Submergence Ratio (*S/D*)

Figure 16 shows the change in flow hydrodynamics in the case of wet bed condition for different still water depths and submergence ratios (*S/D*). Results for all different *d/h* values showed a similar behaviour as in the dry bed case where the water level increased, and flow velocity decreased by the obstruction caused by the presence of the pipe. Figure 16 also shows that in the presence of pipe, the difference between water level and flow velocity with and without pipe is larger in larger still water depths. The root mean square errors (RMSEs) were calculated for flow velocity values and for different still water depths. The calculated RMSE showed a decrease from 0.56 for *d/h* = 0.15, where the pipe is non-submerged (*S/D* = 0, Figure 16e), to 0.19 for *d/h* = 0.425, where the pipe is fully submerged (*S/D* = 1, Figure 16h). RMSE also decreased considerably from 0.40 for the case of less than half submerged pipe (*S/D* = 0.2, Figure 16f) to 0.25 for the case of more than half submerged (*S/D* = 0.6, Figure 16g). Decreased level of pipe submergence resulted in decreased pipe's effective contact area; hence, reduced flow blockage by the pipe. Therefore, the increased level of pipe submergence by increasing still water depth resulted in the reduced influence of pipe presence on flow hydrodynamics. Figure 17 illustrates how the water level rise becomes more abrupt, distinguished with more splashes, at the time of bore impact for cases with lower initial still water depth; thus, smaller level of pipe submergence.

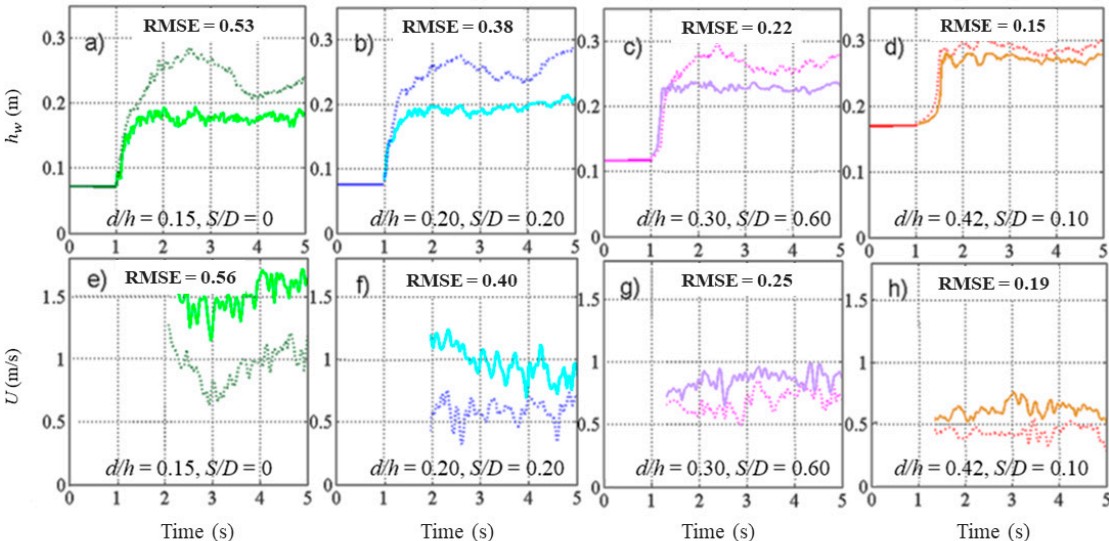

**Figure 16.** Effect of pipe presence on flow hydrodynamics, different still water depth (**d**), wet bed condition and impoundment depth, $h$ = 40 cm, (**a**) and (**d**): $d/h$ = 0.15, non-submerged pipe $S/D$ = 0, (**b**) and (**f**): $d/h$ = 0.2, less than half submerged $S/D$ = 0.2, (**c**) and (**g**): $d/h$ = 0.3, more than half submerged $S/D$ = 0.6, (**d**) and (**h**): $d/h$ = 0.425, fully submerged $S/D$ = 1.0. (Dashed line and continuous line show the "with pipe" and "no pipe" conditions, respectively).

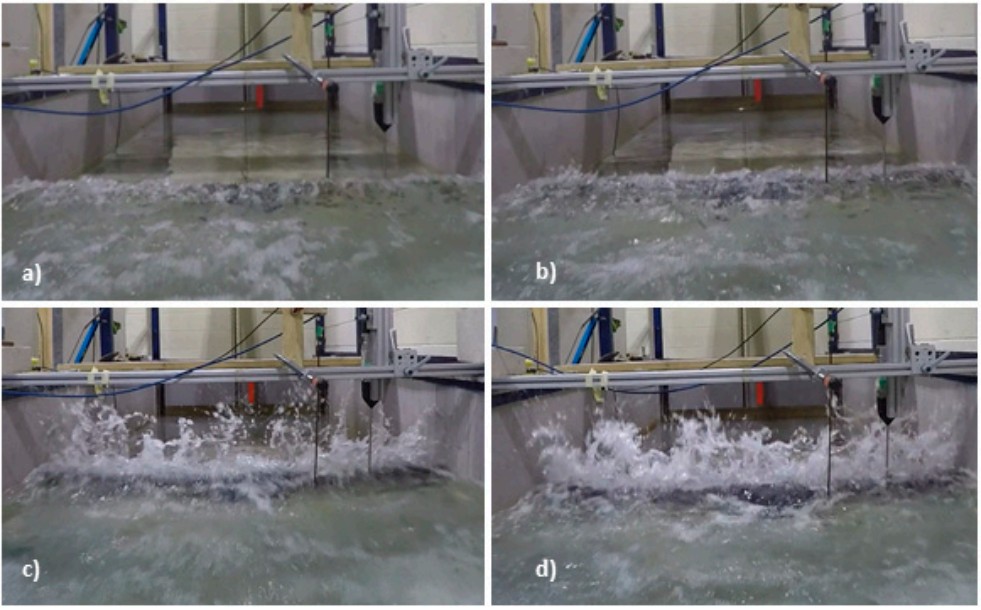

**Figure 17.** Water level rise at the time of bore impact, wet bed condition, and $h$ = 40 impoundment depth cm (**a**) $d/h$ = 0.425, $S/D$ = 1, (**b**) $d/h$ = 0.3, $S/D$ = 0.6, (**c**) $d/h$ = 0.2, $S/D$ = 0.2, (**d**) $d/h$ = 0.2, $S/D$ = 0.

### 3.4. Scale Effects

Table 3 summarizes the flow conditions in this study at the location of the pipe. Previous modelling results from the 1993 Hokkaido-Nansei-Oki Tsunami estimated water depths in the range of 5–15 m and flow velocities in the range of 3–15 m/s. As shown in Table 3, the wave height and wave front celerity measured in this study are in the range of the modelling results mentioned above. According to Lauber and Hager [42], for impoundment depths $h \geq 0.30$ m, inertia forces are dominating and the flow is governed by Froude similarity. This condition applies to all experiments conducted in this study, where the impoundment depths were either equal to or larger than 30 cm.

According to Bricker et al. [44], when using Froude scaling for tsunami modelling, surface tension and viscous effects must be appropriately considered. Weber numbers in Table 3 were calculated using:

$$\text{We} = \frac{\rho u^2 h_w}{\sigma},\tag{8}$$

where $\rho$ is the water density $(\frac{kg}{m^3})$, $u$ is the flow velocity, $h_w$ is the water depth, and $\sigma$ is the surface tension (N/m).

**Table 3.** Hydrodynamic conditions at the pipe location.

| Reservoir Depth $h$ (m) | Head Ratio $d/h$ | Maximum Wave Height (m) | Wave Front Celerity (m/s) | Weber Number (-) | Flow Reynolds Number (-) | Pipe Reynolds Number (-) |
|---|---|---|---|---|---|---|
| | 0.00 | 0.078 | 2.00 | 4285 | $1.56 \times 10^5$ | $2.00 \times 10^5$ |
| | 0.200 | 0.100 | 1.41 | 2730 | $1.41 \times 10^5$ | $1.41 \times 10^5$ |
| 0.3 | 0.260 | 0.107 | 1.53 | 3440 | $1.56 \times 10^5$ | $1.53 \times 10^5$ |
| | 0.400 | 0.136 | 1.35 | 3404 | $1.83\times 10^5$ | $1.35 \times 10^5$ |
| | 0.560 | 0.163 | 1.67 | 6244 | $2.7 \times 10^5$ | $1.67 \times 10^5$ |
| | 0.000 | 0.128 | 2.27 | 9060 | $2.90 \times 10^5$ | $2.26 \times 10^5$ |
| | 0.150 | 0.172 | 1.68 | 6688 | $2.88 \times 10^5$ | $1.68 \times 10^5$ |
| 0.4 | 0.200 | 0.176 | 1.53 | 5660 | $2.69 \times 10^5$ | $1.53 \times 10^5$ |
| | 0.300 | 0.224 | 1.48 | 6739 | $3.31 \times 10^5$ | $1.48 \times 10^5$ |
| | 0.425 | 0.268 | 1.67 | 10,266 | $4.47 \times 10^5$ | $1.67 \times 10^5$ |
| | 0.000 | 0.160 | 2.60 | 14,857 | $4.16 \times 10^5$ | $2.60 \times 10^5$ |
| | 0.120 | 0.215 | 2.05 | 12,411 | $4.40 \times 10^5$ | $2.05 \times 10^5$ |
| 0.5 | 0.160 | 0.220 | 1.85 | 10,342 | $4.07 \times 10^5$ | $1.85 \times 10^5$ |
| | 0.240 | 0.280 | 1.79 | 12,323 | $5.01 \times 10^5$ | $1.79 \times 10^5$ |
| | 0.340 | 0.335 | 1.91 | 16,787 | $6.39 \times 10^5$ | $1.91 \times 10^5$ |

As shown in Table 3, We in all the tested cases are larger than the critical We defined by Peakall and Warburton [45], i.e., We$_{crit}$ ≤ 120. Therefore, the effect of changing surface tension in relation to nature scale could be neglected in this study. According to Te Chow [46], the flow in this study is fully turbulent ($1.41 \times 10^5 < Re < 6.39 \times 10^5$). However, tsunami flow is usually associated with flow Reynolds numbers larger than $10^6$. Therefore, the bottom boundary layer may not be properly represented in the conducted experiments [44]. The present study focuses on the force on the pipe; thus, pipe Re could be more influential. The flow and pipe Reynolds number in the experiments were calculated using:

$$\text{Re} = \frac{h_w u}{\nu},\tag{9}$$

$$Re = \frac{Du}{\nu},\tag{10}$$

where $h_w$ is the water depth, $D$ is the pipe diameter and $\nu$ is the kinematic viscosity. The flow velocity ($u$) in Equation (8) was the free stream velocity measured at the location of the pipe center. Calculated values were in the range of $8 \times 10^4 < \text{Re} < 2.6 \times 10^5$. Wüthrich et al. [35] and Sumer and Fredsøe [47] characterized the flow around the cylinder in this range of Re with a completely turbulent wake and a laminar boundary layer separation both of which cause high pressure and large pressure drag in front of the cylinder. Therefore, turbulent wake flow, which plays an important role on the induced forces on the pipe, is well represented in the experiments.

## 4. Conclusions

The results of this study constitute the first part of a two-part work, presenting an experimental study on the impact of dam-break tsunami-induced hydraulic bores interacting with horizontally-mounted pipelines. The focus of this first part was on the flow hydrodynamics in dry and wet bed conditions and its changing characteristics in the presence of the pipe in the flow. The following conclusions are drawn from this research:

- For the dry bed condition, the bore front celerity increased with an increase in the impoundment depth. $\alpha = 1.2$ was suggested to be used in Equation (4) for the bore front celerity expression.
- The water surface profile and flow velocity, as well as the flow Froude number, were shown to change more gradually over the same period of time for small impoundment depths (i.e., $h = 30$ cm) compared to the waves generated by higher impoundment depths. Momentum flux was also smaller in the wave front region for $h = 30$ cm, due to a smaller flow velocity and water depth.
- Increasing the still water level downstream of the gate led to slower bore flow velocities, reduced Froude number, and reduced momentum flux compared to the bore produced by the same impoundment depth, but propagating over the dry bed. The flow regime changes from supercritical to subcritical with an increase in the still water depth and for $d/h > 0.3$.
- The presence of the pipe, for both dry and wet bed condition, caused the water level to rise and the flow velocity to decrease. In dry bed condition, smaller $e/D$ values resulted in more abrupt water level rise at the time of the bore impact and a faster decrease in flow velocity.
- For bore propagating over dry bed, the water level increase at the time of bore impact in the presence of the pipe became larger with an increasing impoundment depth.
- In the case of the wet bed condition, increased level of pipe submergence $S/D$, due to increasing the still water depth $d$ resulted in a reduction of the influence of the pipe on flow hydrodynamics. This was explained by a reduction in the flow blockage, due to the increased pipe submergence.

**Author Contributions:** B.G. developed the methodology and carried out the experiments. B.G. was also the main responsible for the analysis of the data and writing the manuscript. J.S. assisted in conducting the experiments and contributed in reviewing and editing. I.N. and A.M. conceived of the presented idea, supervised the work and contributed in reviewing and editing. N.G. provided some of the instruments utilized in this study, assisted in the experiments and also contributed in reviewing and editing.

**Funding:** This research was funded by NSERC Discovery Grants held by Ioan Nistor, No. 210282 and Majid Mohammadian, No. 210717. Partial support for the study came through the Marie Curie International Outgoing Fellowship of Nils Goseberg within the 7th European Community Framework Program, No. 622214).

**Acknowledgments:** The authors are grateful to the University of Ottawa Hydraulic Laboratory Technician, Mr. Mark Lapointe, as well as to Adrian Simpalean and Derek Eden, graduate students at the University of Ottawa, for their assistance during the experimental work.

**Conflicts of Interest:** The authors declare no conflict of interest.

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
