# Peer review of "Experimental Study on Extreme Hydrodynamic Loading on Pipelines. Part 1: Flow Hydrodynamics"

_jmse, doi:10.3390/jmse7080251_

Round 1

Reviewer 1 Report

The article presents a thorough characterisation of the flow generated by a dam-break type setup.

It is hard to judge the value of the study without seeing of how exactly will it be used. At present, the article resembles a technical report on flow characterisation.

Figure captions should be expanded to make them self-sufficient.

Figure 15, for example, does not carry any significant information except "yes, there was more splashing". Same for Fig. 17.

70 Hz does not sound a high-speed camera.
lines 210-211: 2 misprints.

Reviewer 2 Report

Ghodoosipour et al. presented an experimental study on the impact of dam-break tsunami-induced hydraulic bores interacting with horizontally-mounted pipelines, paying attention on the flow hydrodynamics in dry and wet bed conditions and its changing characteristics in the presence of the pipe in the flow. The quality of the current experimental work is good and the involved topic is interesting since it investigates problems poor analysed in the literature. Some improvements should be carried out to clarify some parts in order to optimize the overall quality of the paper.All the Reviewer’s requests are listed as follows:

              1.1  Background

·       I suggest to cite an important paper related to the interaction between tsunami-like solitary waves and horizontal circular cylinders: F. Aristodemo, G. Tripepi, D.D. Meringolo, P. Veltri, Solitary wave-induced forces on horizontal circular cylinders: Laboratory experiments and SPH simulations. Coastal Engineering, 129, pp. 17-35, 2017.

2.2.1 Wave gauges

·       Please add the unit to the accuracy of the wave gauges.

·       Define x = 0 the position of the gate.

·       The first wave gauge is WG3 or WG1? Please check.

·       Insert R2and not R2.

2.2.3 Dynamometer

·       Since this work is addressed to understand the hydrodynamics at the cylinder, the information on the dynamometer can be eliminated.

2.3 Experimental program

·      You state that the purpose of the study was to characterize the hydrodynamic forces, but in your present manuscript only the flow hydrodynamic is analysed. Please rewrite this sentence in order to highlight the main aims of the paper.

3.1.1 Dry bed water surface profile

·      Could you quantify the decrease in water level recorded by WG2 and WG3 and the surface roughness of your channel?

·      Correct bd with bed

·      Correct wae with wave

3.1.3 Dry bed flow velocity, Froude number and momentum flux

·      Please insert in Fig. 8 that the colours of the lines refer to h = 30, 40 and 50 cm

3.2.1 Wet bed water surface profile

·      Correct lager with larger

·      A more robust discussion of Fig. 10 about the time feature of dry and wet bed should be incorporated

3.3.2 Wet bed condition

·      Could you quantify the differences in water depth and velocity without and with the pipe?

·      Please introduce in the text the Weber number and the flow Reynolds number

Reviewer 3 Report

The paper presents an experimental study on laboratory simulations of dam-break waves propagating in a rectangular flume, for different initial hydrodynamic conditions, in absence or in presence of a pipe oriented transversely to the flow direction. The paper is the first part of a more complete work, whose goal is the evaluation of the hydrodynamic loading on pipelines produced by extreme flood events generated by tsunamis or hurricanes.

General comments:

The paper essentially shows the variation over time of the main flow variables obtained during the experiments, living to a second paper the discussion on the hydrodynamic effects produced by the dam break-wave on the pipe. In this form, the paper does not show significant elements of originality. Most of the results shown in the paper concern with the difference in water depth or flow velocity obtained for different initial conditions and are in some case obvious. An element of originality should be the description of the effect produced by the dam-break wave on the pipe stability, for different initial position of the pipe and different hydraulic conditions. Some result presented in this paper could be eliminated and a discussion on the stability of the pipe subjected to hydrodynamic forces could be included in the introduction or in another section.      

Specific comments

- In section 2.1, on page 3, line 102, the definition of the non-dimensional gate opening time is not correct:

the ratio between time "t" and the square root of the product between gravity acceleration "g" and the water depth "h" does not produce a non-dimensional quantity. 

- In section 3.1.1, on page 10, line 211, there is a typing error: "wae" should be "wave".

- In section 3.1.3, on page 12, in Figure 8 the legend is missing: the three lines shown in the figure must be explained.

- In the same figure a typing error is present: in the expression of the non-dimensional time, the product inside the root mean square should be a fraction. 

- The same error is present also in Figure 9, on page 13.

- In section 3.3.2, on page 18, line 376, the following sentence is not clear: "in presence of pipe, ....... with and without pipe...".

Reviewer 4 Report

General comments:

This manuscript discusses the tsunami flow hydrodynamics in the presence of horizontally supported pipelines. Tsunamis that occurred in the last decade caused widespread damage to coastal infrastructure and communities. Therefore, studies that attempt to advance the understanding of the tsunami-induced loading on coastal structures (e.g. buildings, bridges, pipelines etc) are relevant and valuable for the development of more resilient communities. Part I of the two-part work does not talk much about the induced loading, which is the main topic of interest for engineers and researchers, however it presents useful information regarding the experimental setup and the data that was recorded. Such data might prove useful for numerical modelers that wish to validate hydrodynamic codes in the future. Therefore, such a manuscript is worthy of publication, provided that the major comments shown below are properly addressed.

Major comments:

1.       Section “1.1. Background”. General comment: The authors reviewed several studies which focused on the tsunami-induced loading on coastal infrastructure including cylinders, buildings and bridges. The authors also talked about some methods that have been used in the past to generate experimentally tsunami-like bores. In order to make the literature review complete, the reviewer would advise the authors to add a new paragraph, where they will talk about different ways to simulate tsunami-like waves. Such a discussion will be very valuable for readers that have no prior experience with tsunamis, and will be also relevant for the field of coastal engineering, where the proper representation of tsunami propagation and loading on structures is currently a topic of interest. For example, previous studies focusing on tsunami-induced loading on coastal infrastructure have used (a) solitary waves, (b) more accurate N-waves, (c) bores generated via the wave-breaking of solitary waves, and (d) dam-break bores. Although solitary waves have been used in the past to study the propagation of tsunami waves and forces on structures, more recent studies (Madsen et al (2008), Chan and Liu (2012), Leschka and Oumeraci (2014), Istrati et al (2018), Zhao et al (2019)), provided evidence that there are significant differences between the forces created by solitary waves and more realistic tsunami-like waves. The authors are advised to take a closer look at the above studies and after summarizing them in a new paragraph, to explain to the reader why they chose to use dam-break bores instead of the other possible ways of representing the tsunami impact on pipelines.

·         Madsen, A., Fuhrman, D.R., Schäffer, H.A., 2008. On the solitary wave paradigm for tsunamis. Journal of Geophysical Research, 113 (C12), 286-292.

·         Chan, I.C., Liu, L.F., 2012. On the run-up of long waves on a plane beach. Journal of Geophysical Research, 117 (C8), 72-82.

·         Leschka, Stefan, and Hocine Oumeraci. (2014). "Solitary waves and bores passing three cylinders-effect of distance and arrangement." Coastal Engineering Proceedings 1, no. 34 (2014): 39.

·         Istrati, D., Buckle, I., Lomonaco, P., & Yim, S. (2018). Deciphering the Tsunami Wave Impact and Associated Connection Forces in Open-Girder Coastal Bridges. Journal of Marine Science and Engineering, 6(4), 148.

·         Zhao, Enjin, Ke Qu, Lin Mu, Simon Kraatz, and Bing Shi. "Numerical Study on the Hydrodynamic Characteristics of Submarine Pipelines under the Impact of Real-World Tsunami-Like Waves." Water 11, no. 2 (2019): 221.

2.       Line 112-113:  How did the authors select the properties of the pipe? Did they adjust the thickness of the pipeline in order to match the stiffness and or strength of the pipeline? Did the authors attempt to model accurately the dynamic properties of the pipelines (e.g. mass, stiffness, period) or do they think such properties would not affect the induced effects on the pipeline? Please provide some additional discussion on this topic.

3.       Line 155-156: The shape of the plates is not clear in Figure 2a. Please add another figure that will show a zoom-in of the plates.

4.       Section 2.2.3: Did the authors record any pressures on the cylinder or just the forces? Understanding the pressure distribution on the pipeline could help understand the underlying physics of the hydrodynamic loading, as well as the localized effects on the pipeline. If the authors did not record any pressures on the cylinder, please explain the reasons for this decision.

5.       Lines 261-266: Indeed the momentum flux is an important parameter that affects the induced loading. However, in order to calculate it the authors used a wave gage and an ADV that have the same distance from the gate but a different in-plane coordinate (=distance from the side walls). Previous research has shown that tsunami-bores can be non-uniform in plane, which means that the free-surface and velocity histories might not be the same at the mid-width of the flume and closer to the walls. Did the authors see any in-plane (looking from the top) non-uniformity of the bore? Some snapshots of the bores before they hit the pipeline (from the high speed camera) would help clarify this issue. Such snapshots would also show if the flow could be assumed to be 2-dimensional (2D) before it impacted the pipeline.

6.       Lines 304-306: The authors make a good point about the influence of the air-entrapment on the velocity measurements. Generally speaking, based on previous experimental studies and the reviewer's experience, the ADV measurements can be quite sensitive to the trapped air and turbulence. What exact criterion did the authors use in order to figure out if the ADV data is valid or not? Did they attempt to compare the ADV data with the velocities calculated by the High Speed Videos (via PIV method)? Such a comparison could increase the confidence in the validity of the experimental data presented herein, improving consequently the value of the current manuscript.

Minor comments:

7.       Section “1.1. Background”: There multiple instances, where the authors use the word “Reference”. For example in line 44 they say “Reference [8]”. It would be more appropriate to replace the word “Reference” with the actual names of the authors. In the above case the authors could say for example “Fritz et al. [8]”.

8.       Lines 210-212: Please replace the words “bd”, “alog” and “wae”, with “bed”, “along” and “wave” respectively.

9.       Lines 248: Too many spaces between the words of the last sentence.

Round 2

Reviewer 4 Report

Manuscript Number: JMSE- 522863

Dear authors,

Thank you for taking the time to address the inquiries of the reviewer. These inquiries have been properly addressed making the paper more comprehensive. Therefore, the manuscript is recommended for publication. I have included only a few minor suggestions, which could be easily addressed by the authors.

Minor Comments:

1.     The authors provided a comprehensive response to the initial inquiry of the reviewer “How did the authors select the properties of the pipe? Did they adjust the thickness of the pipeline in order to match the stiffness and or strength of the pipeline? Did the authors attempt to model accurately the dynamic properties of the pipelines (e.g. mass, stiffness, period) or do they think such properties would not affect the induced effects on the pipeline? Please provide some additional discussion on this topic”.  However, their response has not been included in the manuscript. I think that it would be useful for other readers to see this explanation, so perhaps if the authors find fit, they could summarize their detailed initial response to the reviewer and include a few sentences in the manuscript. For example, they could mention that the pipeline was intentionally designed to be rigid (or to have very high natural frequencies) in order to avoid possible dynamic effects etc. It is up to the authors to decide how much information they will include in the paper.

2.     The authors provided a reasonable response to the initial inquiry of the reviewer ““Indeed the momentum flux is an important parameter that affects the induced loading. However, in order to calculate it the authors used a wave gage and an ADV that have the same distance from the gate but a different in-plane coordinate (=distance from the side walls). Previous research has shown that tsunami-bores can be non-uniform in plane, which means that the free-surface and velocity histories might not be the same at the mid-width of the flume and closer to the walls. Did the authors see any in-plane (looking from the top) non-uniformity of the bore? Some snapshots of the bores before they hit the pipeline (from the high speed camera) would help clarify this issue. Such snapshots would also show if the flow could be assumed to be 2-dimensional (2D) before it impacted the pipeline.” However, their response has not been included in the manuscript. I think that it would be useful for other readers –especially numerical modelers, who might wish to use this data for validation in the future- to see in the paper 2-3 sentences that talk about the fact that the flow is not uniform….so the momentum flux might not be identical across the width of the flume.

3.     At some locations of the paper some sentences use different fonts than the main text, however these errors can be corrected by the authors or the editorial team during the proofreading stage.
